# A global meta-analysis on the drivers of salt marsh planting success and implications for ecosystem services

Zezheng Liu [1], Sergio Fagherazzi [2], Qiang He[3], Olivier Gourgue [4], Junhong Bai [1,5], Xinhui Liu[1,6], Chiyuan Miao [7], Zhan Hu [8,9,10] & Baoshan Cui [1,5,6]

Planting has been widely adopted to battle the loss of salt marshes and to establish living shorelines. However, the drivers of success in salt marsh planting and their ecological effects are poorly understood at the global scale. Here, we assemble a global database, encompassing 22,074 observations reported in 210 studies, to examine the drivers and impacts of salt marsh planting. We show that, on average, 53% of plantings survived globally, and plant survival and growth can be enhanced by careful design of sites, species selection, and novel planted technologies. Planting enhances shoreline protection, primary productivity, soil carbon storage, biodiversity conservation and fishery production (effect sizes = 0.61, 1.55, 0.21, 0.10 and 1.01, respectively), compared with degraded wetlands. However, the ecosystem services of planted marshes, except for shoreline protection, have not yet fully recovered compared with natural wetlands (effect size = −0.25, 95% CI −0.29, −0.22). Fortunately, the levels of most ecological functions related to climate change mitigation and biodiversity increase with plantation age when compared with natural wetlands, and achieve equivalence to natural wetlands after 5–25 years. Overall, our results suggest that salt marsh planting could be used as a strategy to enhance shoreline protection, biodiversity conservation and carbon sequestration.

Salt marshes are vegetated ecosystems located between the sea and land, provide many valuable ecosystem services, and form a sustainable nature-based coastal protection[1–3]. The feedbacks between vegetation and geomorphology amplify carbon sequestration and storage, so that salt marshes rank among the most effective carbon sinks on the planet, delivering significant capacity for climate change mitigation and adaption[4–7]. Additionally, salt marshes can trap large amounts of sediments and dissipate wave energy, protecting humans

[1]State Key Laboratory of Water Environmental Simulation, School of Environment, Beijing Normal University, Beijing 100875, China. [2]Department of Earth and Environment, Boston University, Massachusetts 02215, USA. [3]Coastal Ecology Lab, MOE Key Laboratory for Biodiversity Science and Ecological Engineering, School of Life Sciences, Fudan University, Shanghai 200438, China. [4]Operational Directorate Natural Environment, Royal Belgian Institute of Natural Sciences, 1000 Brussels, Belgium. [5]Yellow River Estuary Wetland Ecosystem Observation and Research Station, Ministry of Education, Shandong 257500, China. [6]Research and Development Center for Watershed Environmental Eco-Engineering, Beijing Normal University at Zhuhai, Zhuhai 519087, China. [7]State Key Laboratory of Earth Surface Processes and Resource Ecology, Faculty of Geographical Science, Beijing Normal University, Beijing 100875, China. [8]School of Marine Sciences, Sun Yat-Sen University, and Southern Marine Science and Engineering Guangdong Laboratory (Zhuhai), Zhuhai 519000, China. [9]Guangdong Provincial Key Laboratory of Marine Resources and Coastal Engineering, Guangzhou 510000, China. [10]Pearl River Estuary Marine Ecosystem Research Station, Ministry of Education, Zhuhai 519000, China. ✉e-mail: huzh9@mail.sysu.edu.cn; cuibs@bnu.edu.cn

and infrastructure against sea level rise and storm surges[8,9]. Salt marshes host unique ecosystems that provide habitat for fish and birds and support high biological productivity[1]. However, these vegetated ecosystems have been lost or severely degraded around the world primarily due to anthropogenic disturbances[10,11], resulting in the loss of ecosystem services and the reduction in ecological functions[12–14].

To halt and reverse the degradation of salt marshes, numerous efforts have been carried out globally to conserve and restore these coastal vegetated ecosystems and provide nature-based solutions to mitigate climate change[15–17]. Planting or revegetation in salt marshes is the most traditional and popular restoration strategy, and has been used for shoreline protection, climate change mitigation and adaptation, or habitat development[17–19]. To our knowledge, the earliest documented example of planted *Spartina alterniflora* Loisel. for erosion control in the United States dates back to the 1920s[20]. In recent years, to implement global agendas for climate mitigation, biodiversity conservation and sustainable development, such as the Paris Agreement[21], the United Nations Decade on Ecosystem Restoration[22] and the UN Sustainable Development Goals[23], many countries are developing national and regional coastal restoration plans and policies, in which efforts to overcome coastal degradation involve salt marshes planting[18,23]. However, even traditional methods of marsh transplanting can be risky: where species, site selection, or planting technologies are inappropriate, salt marsh restoration can fail[17,24,25]. Moreover, the potential of marsh transplanting to provide carbon-related benefits, improve ecological functioning and enhance biodiversity is highly variable[26,27].

It is clear that planting in salt marshes is often challenging, with many projects exhibiting low establishment success and high mortality[28,29]. Thus, there is an urgent need to understand what elements contribute to overall planting success. In recent years, the attention has shifted to restoring key ecological functions, including shoreline protection[30], carbon sequestration[31,32], biodiversity conservation[33] and nursery for fisheries[34,35]. Previous studies on the ecological outcomes of planted marshes were mostly narrative syntheses of individual projects around the world[23,36–38]. A few quantitative studies were limited to specific areas or functional categories, and show divergent trends because of the dynamic nature of salt marshes and the distinct context of each location[39,40]. This limits prediction of restoration outcomes and effective resource management[26]. A large quantitative synthesis provides an opportunity for a systematic assessment of the effects of planting in different regions of the world and the mechanisms that underpin them, as well as helps identify research trends and gaps to guide effective strategic policy and action for biodiversity conservation and climate change mitigation[41–43].

Here we present a global synthesis from 210 publications of planting outcomes in salt marshes. We quantify the survival and growth of outplants as success indicators and how there are affected by abiotic and biotic filters. We also examine the effects of salt marsh planting on a range of ecological effects related to climate change mitigation and biodiversity conservation, and compare the outcomes to natural and degraded wetlands. For each variable related to shoreline protection, primary productivity, soil carbon storage, greenhouse gas (GHG) fluxes, biodiversity conservation and fishery production, we conduct a meta-analysis to quantify both log response ratios and Hedges' $g^*$ effect sizes, as a means of providing a quantitative estimate of salt marsh planting performance. Then, we examine how the effect sizes of planting varied with time since planting. For all analyses, we test for publication bias and robustness using the funnel plot and Rosenthal's fail-safe number. Our results provide evidence for the potential of salt marsh planting to support biodiversity conservation and climate change mitigation and have implications for restoration policy and predictive restoration models.

## Results

### Global patterns of salt marsh planting

Planting in salt marshes has been widely used around the world. Our study spans a total of 15 countries on five continents along the shorelines of the Atlantic, Pacific, and Indian Oceans. Most studies are in North America (168 studies), Europe (24 studies) and East Asia (16 studies) (Fig. 1a). Over 90 plant species encompassing 45 genera were utilized in salt marsh planting restoration efforts. Salt marshes were mostly planted with one species or genus (72.4% of all studies), rather than two or more genera (27.6%). The most commonly planted genus were *Spartina* (76.2%), *Juncus* (11.4%), *Scirpus* (8.1), *Salicornia* (5.7%) and *Suaeda* (5.2%) (Fig. 1a; Supplementary Table 1). For all studies, plugs, shoots, clumps, or patches (128 studies) were the most common material planted, whereas only 13 studies used direct seeding in restoration projects.

Often planting in salt marshes is somewhat of a gamble. Among all observations, the mean survival rate was only $53 \pm 37$ % ($n = 1038$) (Fig. 1b). There was no statistically significant difference between restoration projects and experimental manipulations of plantings ($P = 0.557$) (Supplementary Fig. 2). Frequency distribution showed that the observations of survival did not follow a normal distribution ($P < 0.001$) (Fig. 1b, Supplementary Figs. 3–5). We found evidence that a portion of restoration efforts had a high survival rate. For instance, in 17% of observations, more than 95% of planted transplants survived during monitoring. However, less than 5% planted transplants survived in 23% of observations (Fig. 1b). We also found that survival of transplants varied significantly among genera ($P < 0.05$), and several genera (e.g. *Puccinellia* 79.31%, *Juncus* 76.97%, *Batis* 76.04%, *Salicornia* 75.03%, *Spartina* 73.31% and *Frankenia* 71.55%) have higher average survival than others (e.g. *Carex* 48.33%, *Phragmites* 33.65%, *Suaeda* 26.29% and *Scirpus* 20.86%) (Fig. 1c).

### Constraints on the fitness of outplants

Across the entire data set, a series of abiotic and biotic factors are critical determinants of transplant's survival and growth, reflecting different aspects of potential ecological filters (Fig. 2a). First, intrinsic species characteristics such as taxonomic grouping, propagule types and plant source are the primary determinants for successful restoration. These reflect differential plant traits and tolerance thresholds that influence plant growth and survival. Second, ecological assembly filters have most commonly focused on abiotic constraints (71 studies), which are closely linked to tidal flat elevation and resource availability (Fig. 2a). Successful planting in salt marshes requires passing these physical thresholds and open windows of opportunity for establishment and persistence of target species[44]; in contrast, unsuitable planting time, site conditions (that is, high erosion or sediment deposition, hypersalinity and nutrient stresses) depressed the probability of restored species success (Supplementary Table 2; Fig. 2d). Third, several prominent biotic filters (35 studies) were considered: facilitation, competition and herbivory (Fig. 2a). Management of facilitation and competition in planting commonly focuses on controlling species richness, spacing and density of planting seedlings. In line with the stress-gradient hypothesis, these positive/negative inter- and intraspecific interactions can shift under different environmental stress levels and species' traits. The effect of herbivores, including snails, crabs and waterfowl, on the survival of planting in salt marshes were observed in few studies (Supplementary Table 2).

Given these overall patterns, we then explored the drivers that could characterize the presence of a target species after planting efforts. As a result, high planting density, tight planting spacing, and large planting size have an overall positive influence on species success, which significantly increased plant survival, density, height, and plant above- and belowground biomass (Fig. 2f). Locally collected plants had higher survivorship ($n = 43$, RR' = 0.16, 95% CI 0.09, 0.23, $P = 0.0001$) than plants from non-local sources, although their effects

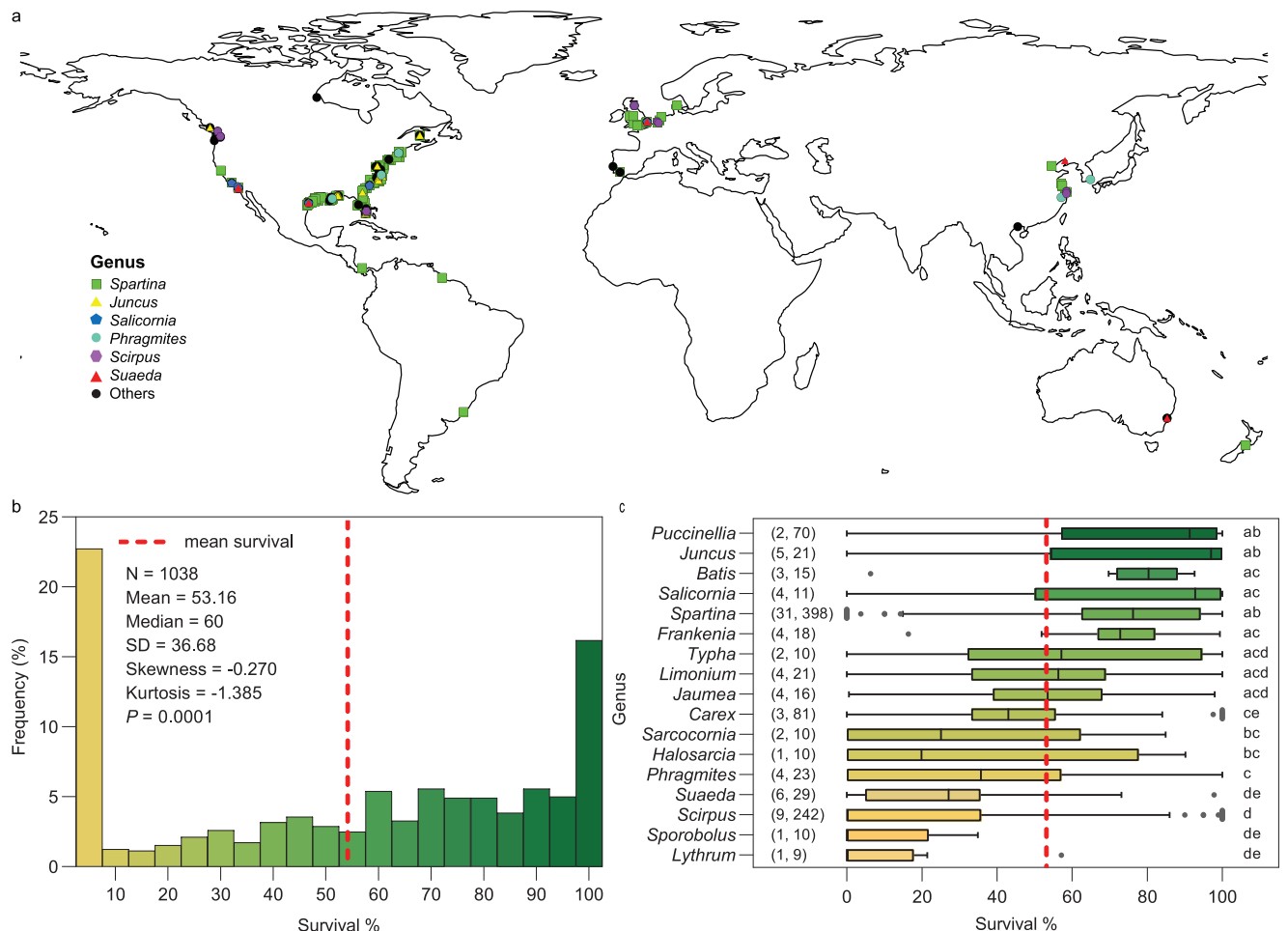

**Fig. 1 | Global distribution and survival of planting in salt marshes.**
**a** Geographical location of the 210 studies included in this synthesis. Global coastline data is available at Natural Earth (naturalearthdata.com), which provides free vector and raster map data. Global map was created using ArcGIS 10.2 software. **b** Frequency distribution of survival; numbers in the graph show basic statistical information. Normality of distribution was tested for skewness and kurtosis with a Kolmogorov–Smirnov test using GraphPad Prism 8 (v8.00). *P*-values reflect two-sided tests. **c** Survival of each genus; range of survival for each plant genus represented as box and whisker plots with quartiles, median, and outliers. Central lines in the boxes represent medians, left edges of the boxes represent first quartiles, right edges of the boxes represent third quartiles, whiskers indicate 1.5 times the interquartile distance, and individual points are outliers. The first and second numbers in parentheses indicate, respectively, the number of studies and observations included in each genus. Differences in survival between genera were assessed using the non-parametric, two-sided Wilcox test, performing with GraphPad Prism 8 (v8.00). Different letters denote significantly different (*P* < 0.05). The exact *P*-values are presented in Supplementary Table 4. Source data are provided as a Source Data file.

on plant density and height were insignificant (Fig. 2f, Supplementary Table 6). Artificially augmented plant richness increased plant survival (*n* = 24, RR′ = 0.12, 95% CI 0.07, 0.18, *P* = 0.0001) and belowground biomass (*n* = 4, RR′ = 0.53, 95% CI 0.27, 0.79, *P* = 0.0001), but other canopy structures (density, height, coverage) and total aboveground biomass were largely unaffected by mixed planting treatments (Fig. 2f, Supplementary Table 6). Fertilizer treatments with nitrogen or phosphorus addition significantly increased plant survival (*n* = 20, RR′= 0.11, 95% CI 0.04, 0.19, *P* = 0.004), height (*n* = 15, RR′ = 0.19, 95% CI 0.09, 0.30, *P* = 0.0003), coverage (*n* = 6, RR′ = 0.60, 95% CI 0.38, 0.82, *P* = 0.0001), expansion (*n* = 32, Hedges' *g*\* = 0.51, 95% CI 0.37, 0.65, *P* = 0.0001) and plant aboveground biomass (*n* = 34, RR′ = 0.97, 95% CI 0.58, 1.36, *P* = 0.0001), but not density and belowground biomass (Fig. 2f, Supplementary Table 6). Protective structures, including wire netting with mesh to protect plantings from grazing and artificially trait-mimicry structures that reduce stem movement and stabilize sediment (Supplementary Table 3), significantly increased plant survival (*n* = 3, RR′ = 0.57, 95% CI 0.16, 0.98, *P* = 0.0068), density (*n* = 22, RR′ = 0.32, 95% CI 0.16, 0.48, *P* = 0.0001) and aboveground biomass (*n* = 25, RR′ = 0.21, 95% CI 0.14, 0.28, *P* = 0.0001), but not height and

belowground biomass (Fig. 2f, Supplementary Table 6). In addition, direct seeding (Fig. 2b), exposed wave forcing conditions (Fig. 2d) and herbivory (Fig. 2e) negatively influenced the probability of planting success; the spring/summer transplants had much better survival than the autumn/winter ones (Fig. 2c).

## Ecological effects of planting

Compared with natural marshes, planted marshes have a lower ability to deliver different functions, except for shoreline protection (97 studies) (Fig. 3a and Supplementary Data 1). Planting in salt marshes significantly increased accretion and elevation change rates by 13.32 ± 4.60 and 26.10 ± 5.88 mm yr⁻¹ (*n* = 88 and 93, respectively), and might increase sediment trapping by on average 41.76 ± 16.2 ton ha⁻¹ yr⁻¹ (*n* = 44) with respect to adjacent natural wetlands (Supplementary Data 1). Planted marshes provide lower levels of primary productivity and soil carbon storage compared with natural wetlands. Belowground biomass (*n* = 317) and subsurface soil carbon content/density (>10 cm deep) (*n* = 99 and 34, respectively) develop more slowly than aboveground biomass (*n* = 359) and surface soils (0–10 cm deep) (*n* = 136 and 14, respectively) (Fig. 3a). However, the level of soil organic carbon

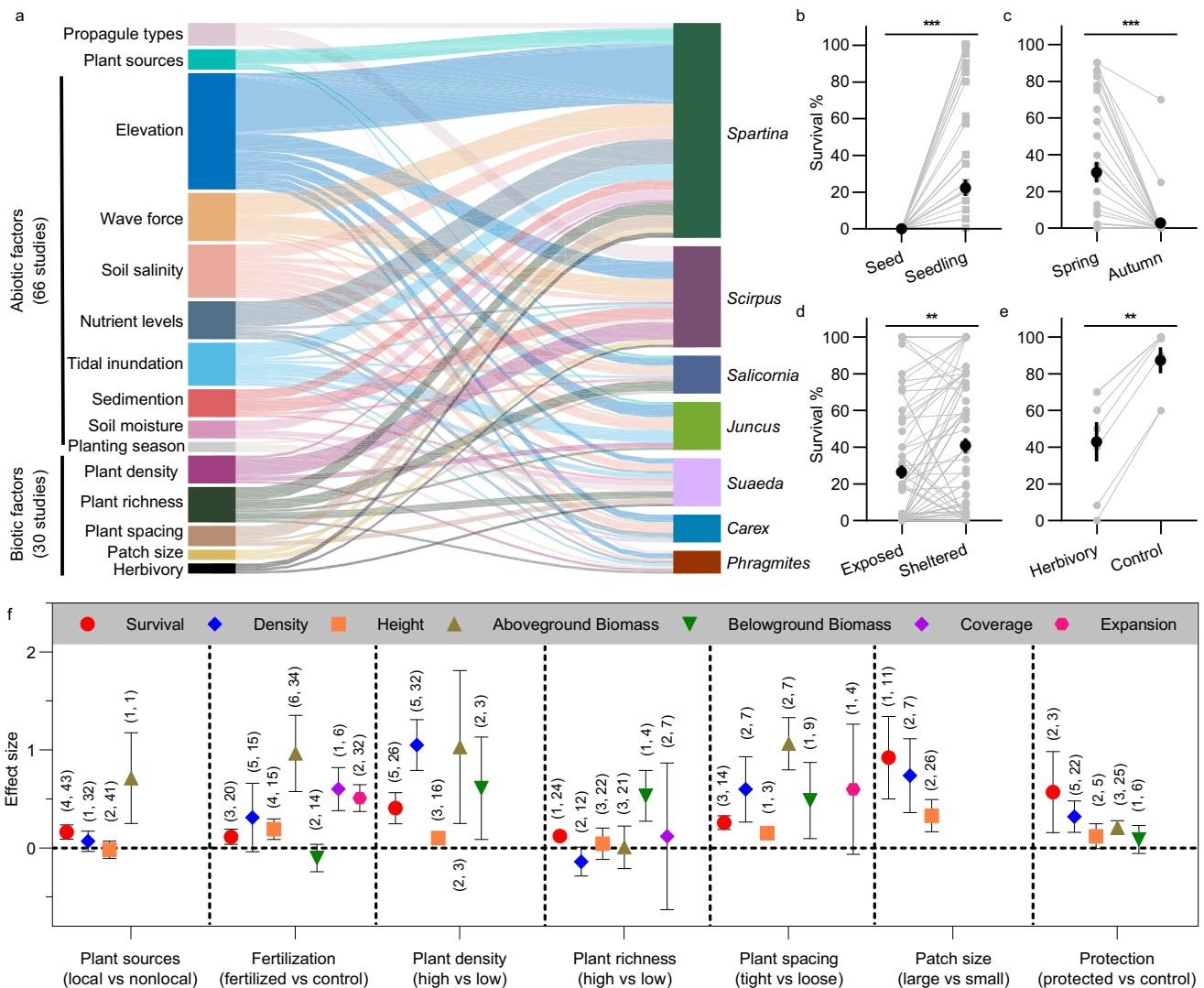

**Fig. 2 | Constraints on planting establishment and the effect size of plant performances. a** Planting establishment is influenced by ecological filters and species characteristics including a series of abiotic and biotic factors. Sankey diagram in which the thickness of the lines between the left and right columns represents the number of studies between the factors and plant genus they involved. **b–e** Plant survival rates under different propagule types (**b**), planting season (**c**), wave force (**d**) and herbivory (**e**) (sample sizes = 61, 37, 85 and 7, respectively). Each line represents the difference in survival rate in a planted with seed and seedlings pair (**b**), planted in spring and autumn season pair (**c**), planted in the wave-exposed site and sheltered site pair (**d**), plant in an herbivory and control treatment with grazing exclusion pair (**e**). In panel (**b–e**) dots represent means and vertical lines show standard errors. The differences in survival rates were analysed using the Kruskal–Wallis *H* test using GraphPad Prism 8 (v8.00). The exact *P*-values are 0.000002, 0.000014, 0.008 and 0.0049, respectively. Asterisks indicate the significance: **P < 0.01; ***P < 0.001. **f** Overall effect size of factors selected on different plant performance measures in planting efforts. Shown are effect sizes in mean and 95% CIs. Effect sizes are considered significant if their 95% CI does not overlap zero. The first and second numbers in parentheses indicate the number of studies and observations included in each calculation, respectively. Source data are provided as a Source Data file.

(SOC) accumulation rate was not different between restored ($64.26 \pm 7.54$ g/m²/year) and natural ($64.72 \pm 9.31$ g/m²/year) marshes (n = 18) (Supplementary Data 1). In addition, there were no significant differences in $CO_2$ (n = 86, Hedges' $g^* = 0.03$, 95% CI −0.10, 0.17, P = 0.62) and $N_2O$ (n = 70, Hedges' $g^* = 0.12$, 95% CI −0.03, 0.27, P = 0.12) fluxes between restored and natural marshes, and restored marshes have significantly lower $CH_4$ flux (n = 70, Hedges' $g^* = -0.21$, 95% CI −0.38, −0.03, P = 0.02) than their natural sites (Fig. 3a, Supplementary Table 7). These suggested that restored marshes have the potential to equal or exceed the carbon sequestration capacity of the natural marshes. Furthermore, the levels of biodiversity of vegetation and macrobenthos are much lower for restored marshes compared to natural marshes. For instance, restored marshes have lower levels of species richness for vegetation (n = 26, RR' = −0.41, 95% CI −0.57, −0.24, P = 0.0001) and macrobenthos (n = 64, RR' = −0.23, 95% CI −0.34,

−0.12, P = 0.0001), and less abundance and biomass of macrobenthos (428 and 62, respectively) than natural marshes (Fig. 3a, Supplementary Table 7). Similarly, planted sites have less size, biomass and abundance of fishery species than natural marshes (84, 64 and 746, respectively) (Fig. 3a and Supplementary Data 1). However, this is not evident for all individual functions, for example, planted marshes tended to support higher abundance of birds (n = 27, RR' = 0.64, 95% CI 0.08, 1.20, P = 0.0247) and catch rate of fishery species (n = 147, RR' = 0.50, 95% CI 0.34, 0.65, P = 0.0001) than nearby natural marshes (Fig. 3a, Supplementary Table 7). Thus, when compared with natural marshes, the meta-analysis indicates that planting in salt marshes could play a limited but promising role in climate change mitigation and biodiversity conservation, especially in controlling coastal erosion under rapid sea level rise and enhancing the abundance and diversity of avian and fishery communities.

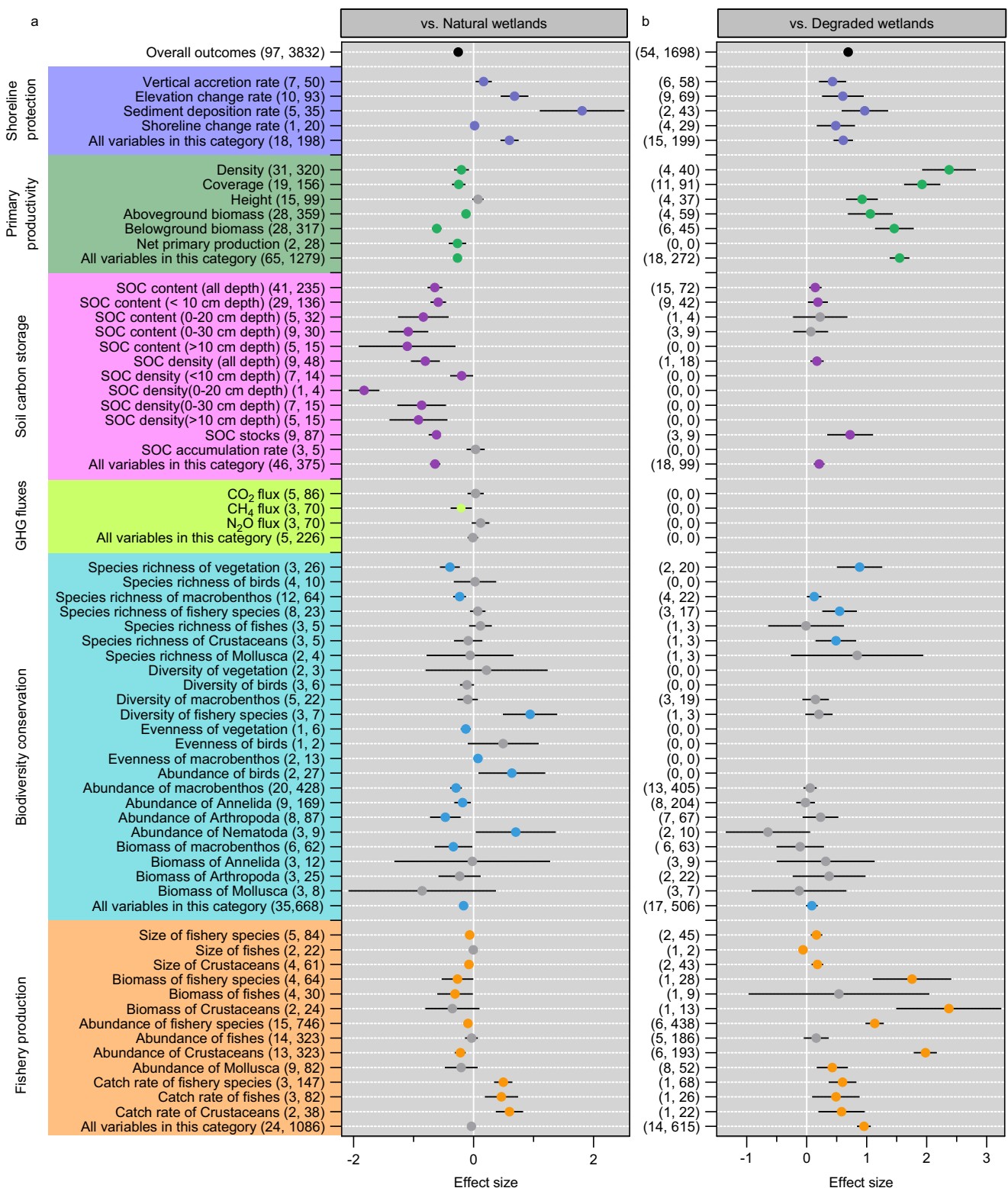

**Fig. 3 | Meta-analysis of ecological effects produced by planting in salt marshes.** **a** Effect sizes of planted salt marshes compared with natural wetlands. **b** Effect sizes of planted salt marshes compared with degraded wetlands. Shown are mean effect sizes with 95% CIs. Effect sizes are considered significant if their 95% CI does not overlap zero. Effect sizes without a significant trend are shown in gray. The first and second numbers in parentheses represent the number of studies and observations. SOC soil organic carbon, GHG greenhouse gas. Source data are provided as a Source Data file.

Compared to degraded marshes, however, planted marshes have better restoration outcomes (54 studies) (Fig. 3b, Supplementary Data 1 and Supplementary Table 8). Restored marshes exhibit higher levels of sediment accretion, primary productivity, and soil carbon

sequestration compared with degraded marshes. For instance, planting helps dissipate wave energy causing shoreline erosion, and convert eroding into depositional environments producing shore progradation. Planting in salt marshes significantly increased aboveground and

belowground biomass by 237.88 ± 30.53 and 479.18 ± 145.30 g/m² (n = 59 and 73, respectively), and increased SOC stocks by on average 10.70 ± 3.45 Mg C/ha (n = 9) with respect to adjacent degraded wetlands (Supplementary Data 1). Furthermore, planted marshes have higher species richness of vegetation, macrobenthos and fishery species than degraded marshes (n = 20, 22 and 17, respectively). The size, biomass, abundance and catch rate of fishery species are much higher for restored marshes compared with degraded marshes (n = 45, 28, 438 and 68, respectively). However, there was no major difference for other functions such as abundance and biomass of macrobenthos (n = 405 and 63, respectively) (Fig. 3b and Supplementary Data 1). To some degree, the outcomes of our meta-analysis for these ecological functions should be interpreted with caution, because the number of studies containing matched pairs of restored and degraded marshes was quite limited. Overall, when compared with degraded wetlands, the meta-analysis indicates that planting in salt marshes provides a promising option to mitigate climate change and to improve biodiversity.

### Effects of plantation age

Marshes age after planting ranged between 1 and 36 years old in the reviewed cases, with most studies reporting the ecological effects in the first decade (Supplementary Fig. 6). The meta-regression revealed that most ecological effects increased logarithmically with age when compared with natural wetlands (Fig. 4a; Supplementary Table 10). The effects of plantation on primary productivity and SOC storage increased with plantation age for all individual functions, except for SOC content (>10 cm) (Fig. 4a). Specifically, increasing age significantly improved performance of planted marshes relative to that of native wetlands with respect to plant density, coverage, belowground biomass, SOC density (<10 cm), SOC stocks, abundance of Mollusca, species richness and abundance of macrobenthos (Fig. 4b, c). Plant belowground biomass and SOC stocks required longer time (over 25 years) than plant density to reach the similar levels that in natural marshes (Fig. 4b). Conversely, increasing age significantly reduced the performance for sediment deposition rates, richness of bird species and catch rate of fishery species (Fig. 4b, c), indicating that young restored marshes may provide better nursery and habitat to bird and fishery species than older restored marshes.

Compared to degraded wetlands, several ecological effects decreased logarithmically with planted age, particularly most individual functions in primary productivity and SOC storage (Fig. 4a). Along a gradient of plantation age, plant height, aboveground biomass, SOC content (0–30 cm), species richness of plant showed significantly decreasing trends (Fig. 4d, e). Once the vegetation community became established, the restored marshes had greater plant height, aboveground biomass, and species richness of plant than the degraded marshes (Fig. 4d, e). However, SOC stocks exhibited a significant

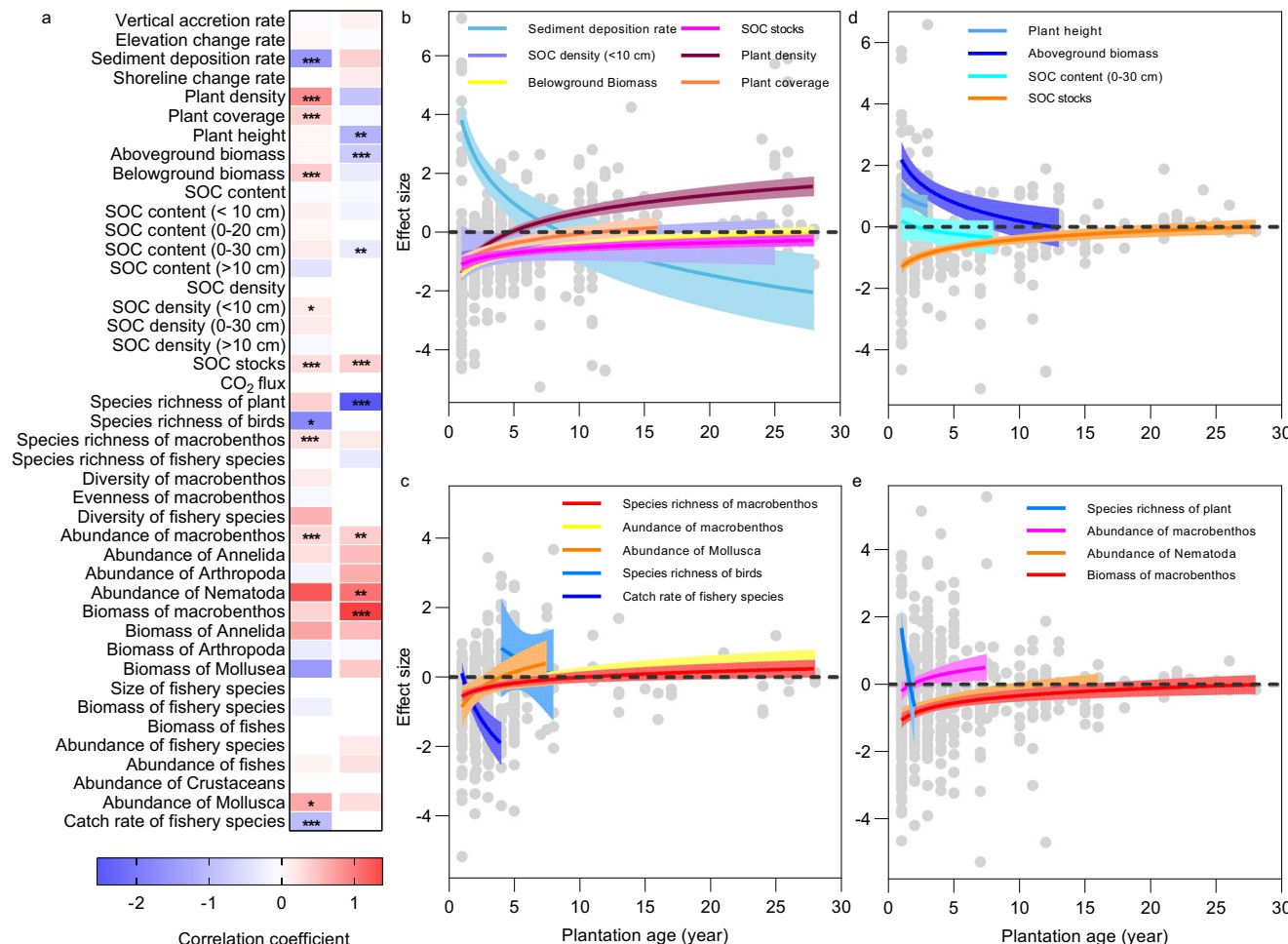

**Fig. 4 | Effects of restored age on ecological outcomes. a** Heatmap presents the matrix of correlation coefficients between ecological outcomes and planted age. The red and blue color indicate positive and negative effects of age, respectively. P-values are predictions from the meta-regression model, and reflect two-sided tests. Asterisks indicate the significance: *P < 0.05; **P < 0.01; ***P < 0.001. **b–e** Relationship between plantation age and the ecological effects in comparison with natural wetlands (**b**, **c**) and degraded wetlands (**d**, **e**). The regression lines and 95% CIs are predictions from the meta-regression models. Dashed horizontal lines indicate zero. Source data are provided as a Source Data file.

increase pattern when comparing restored marshes with degraded marshes (Fig. 4d). Furthermore, the most effect sizes for biodiversity conservation and fishery production showed increasing trends along a gradient of plantation age (Fig. 4a). Specifically, abundance of Nematoda, abundance and biomass of macrobenthos significantly logarithmically increased with plantation age compared with degraded wetlands (Fig. 4d, e). Generally, abundance of Nematoda and biomass of macrobenthos achieved equivalence to degraded wetlands after 10–25 years (Fig. 4e). We also found no significant relationships between effect sizes and plantation age for some functions, such as vertical accretion and elevation change rates, $CO_2$ flux (Fig. 4a).

## Discussion

This systematic review with meta-analysis suggests that salt marsh planting is a promising strategy for biodiversity conservation and climate change mitigation and adaption. Specifically, we find that mean survival is 53% for transplants, limited by intrinsic species characteristics and a series of abiotic and biotic filters. However, careful design of sites, species selection, and novel planted technologies can facilitate the survival and establishment of planted organisms. Additionally, our study shows that planting enhanced carbon sequestration, biodiversity, and shoreline protection as compared with degraded wetlands. Compared with natural wetlands, however, the ecosystem services of planted marshes, except for shoreline protection, have not yet fully recovered. Fortunately, most effects of planting on carbon storage and biodiversity conservation increased with planted age. Thus, salt marsh planting contributes to the implementation of global agendas for climate change adaptation and mitigation, biodiversity conservation and sustainable development.

Survival of transplants is one of the restoration success indicators, because the potential of planting to provide restoration benefits depend on plant survival, establishment and expansion. When planting efforts were successful, established vegetation could dissipate wave energy, enhance sedimentation, store carbon and support biodiversity conservation[4,6,45]. Planting failure (survival rate ≤10%) occurred for a number of reasons (Supplementary Table 2). Planted transplants were particularly sensitive to the species used and multiple stresses including high erosion or sediment deposition, nutrient stresses and animal activity (e.g., grazing) (Supplementary Table 2). Planting projects or field experiments to improve survival and growth used a range of techniques, including planting in clumps or at high density[46,47], transplanting cores or plugs[28], transplanting within structures mimicking emergent traits[48], co-transplanting with mussels[49], species-rich planting[50], planting with fertilization[51], among others (Fig. 2f; Supplementary Table 3). In the future, thus, careful design of site and species selection and new planting technologies are required to open the windows of opportunity for target species establishment[44], and facilitate long-term success of planted organisms.

The potential of planting to provide restoration benefits also depend on restoration age, and some selected ecological effects increased with planted age (Fig. 4a). For instance, the effect size of primary productivity and soil carbon storage compared with natural wetlands increased with planted age (Fig. 4a, b), indicating that planting in salt marshes restored the capacity of the ecosystem to store carbon and help rebuild the lost carbon sink. The effect sizes of macrobenthos species richness and abundance are also strongly related to planted age, which provides evidence of a clear benefit of planting for biodiversity conservation (Fig. 4a, b). These results are consistent with those reported in individual case studies from different parts of the world[27,52]. However, several restoration benefits are not sensitive to planted age, such as species richness of overall fishery species and diversity of macrobenthos. These processes may be more sensitive to other factors such as hydrographic conditions, food resources and refuge structures among habitats[45,53,54]. Unsurprisingly, several restoration benefits somewhat slowly developed, and did not achieve

equivalence to natural undisturbed marshes even after decades[27,55,56]. Thus, the potential of salt marsh planting for climate change mitigation and biodiversity conservation may require many decades or centuries to develop[57].

Individual studies also identify numerous environmental factors that affect the ecological effects of salt marsh restoration, such as surface elevation, sediment and water characteristics, and variable disturbance histories. Planted salt marshes have always lower elevation than natural marshes (Fig. 5a; Supplementary Table 5), possibly as a result of erosion, compaction and decomposition of dead belowground organic matter[58]. The lower initial elevations of planted marshes potentially allow for longer flooded periods, greater sedimentation and elevation increase than in the natural marshes[40,58] (Fig. 3a). Furthermore, planted marsh soils were dominated by sand, whereas natural marshes contained less sand and more silt and clay, resulting in higher soil redox potential (Eh) in the soils (Fig. 5a; Supplementary Table 9). The sandy textures always cause organic matter to decompose faster than natural marsh soils, likely reducing the accumulation of soil organic carbon[27,59,60]. Due to limited organic matter, planted marsh soils have high soil bulk density (Fig. 5a). The higher bulk density and soil salinity, combined to lower soil moisture may adversely affect the successful establishment of transplants and the ability to provide hydrologic and nutrient cycling functions[24,61,62]. Conversely, there are no difference in surface water characteristics (e.g. surface water salinity, dissolved oxygen, temperature and turbidity) between planted and natural marshes (Fig. 5a). On the other hand, there are few differences in environmental factors between planted and degraded marshes (Fig. 5b). Surface elevation and soil grain size were comparable to those in degraded wetlands (Fig. 5b; Supplementary Table 5), indicating that restored marshes had similar hydrologic and geomorphic regimes.

For biodiversity conservation, our findings suggest relatively more rapid development of avian and nekton communities in planted marshes than the benthic invertebrate community (Fig. 3a), perhaps because the highly mobile nature of birds and fish facilitates rapid colonization of restored habitats[45,63]. Planted marshes may also increase the structural complexity of the habitat and attract a diverse assemblage of shorebirds and nekton[45,64]. Planted marshes also provide better refuge from predation and environmental stress and higher food resource availability for shorebirds and nekton[45,63,65,66]. The effect sizes of species richness of birds and catch rate of fishery species significantly decreased with constructed marsh age, whereas species richness and abundance of macrobenthos increased with marsh age (Fig. 4c, d). Their distinct development trajectories, initially rapid then decreasing gradually over time for avian and nekton communities and increasing gradually over time for benthic invertebrate communities, may be due to the a trend of increasing similarity between planted and natural marshes with age[27,67,68].

Almost inevitably, cross-study syntheses are prone to publication bias by comparing across multiple spatial scales or replication units[43,69]. The funnel plot and Rosenthal's fail-safe number suggest that our meta-analysis was generally robust with respect to publication bias, except for few effect sizes with a small sample size (Supplementary Figs. 7–10 and Supplementary Table 6–9). For instance, the funnel plots suggest that for plant survival and growth, there is no publication bias, except for the mean effect sizes of plant density on belowground biomass, plant richness and fertilization on plant density (Supplementary Fig. 7 and Supplementary Table 6). Similarly, after adjusting any detected potential publication bias using the "trimfill" method, we find that the most effect sizes of ecological outcomes did not change the magnitude and direction of effect sizes and their 95% CIs (Supplementary Figs. 8–10 and Supplementary Tables 7–9). When the potential publication bias was adjusted, the mean effect sizes of vertical accretion rate, SOC content (>10 cm), SOC density (<10 cm), abundance of Arthropoda and biomass of macrobenthos as compared

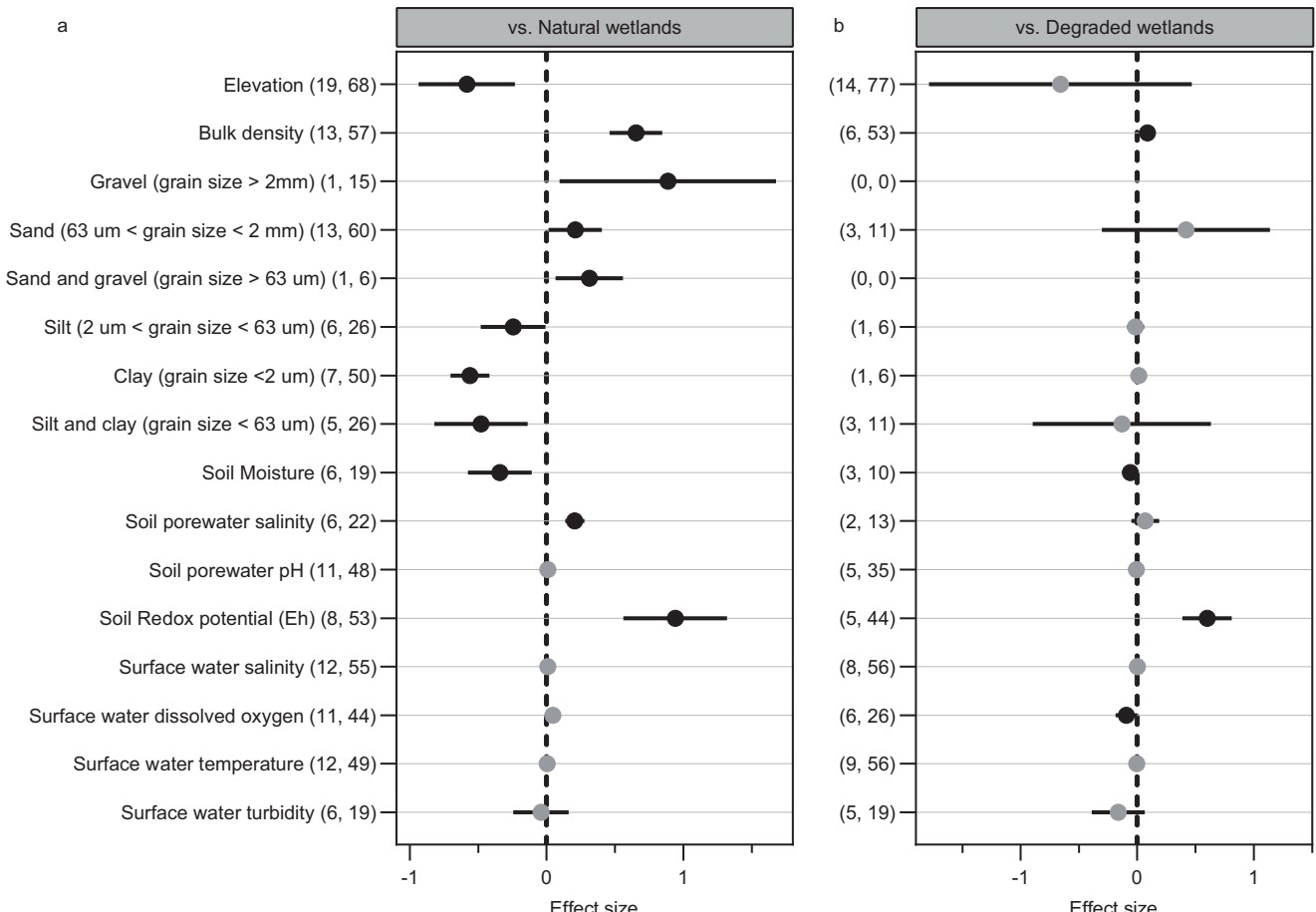

**Fig. 5 | Difference of environmental factors in restored, natural and degraded salt marshes. a** Effect sizes of planted salt marshes compared with natural wetlands, **b** Effect sizes of planted salt marshes compared with degraded wetlands. Shown are mean effect sizes with 95% CIs. Effect sizes are considered significant if their 95% CI does not overlap zero. Effect sizes without a significant trend are shown in gray. The first and second numbers in parentheses indicate the number of studies and observations, respectively. Source data are provided as a Source Data file.

with natural wetlands, and shoreline change rate as compared with degraded wetlands became insignificant (Supplementary Figs. 8–9 and Supplementary Tables 7–8). In some cases (e.g. SOC content, SOC density, abundance of Arthropoda and shoreline change rate), however, Rosenthal's fail-safe number was much greater than $5n + 10$, indicating that these mean effect sizes are robust to publication bias (Supplementary Tables 6–9).

Finally, this systematic review with meta-analysis has major implications for designing appropriate evidence-based restoration plans and policies, and predictive restoration models. Our results suggest that planted marshes provide ecological effects (e.g. shoreline protection, primary productivity, soil carbon storage, biodiversity conservation and fishery production) at higher levels than degraded wetlands. This points to the potential of salt marsh planting to contribute meaningfully to the UN Decade on Ecosystem Restoration (2021–2030), the UN Decade of Ocean Science for Sustainable Development (2021–2030), post-2020 Global Biodiversity Framework, the Paris Agreement and the UN Sustainable Development Goal. However, the levels of most ecological effects, except for shoreline protection, are generally lower for restored marshes compared to natural marshes. Thus, natural marshes are not replaceable in the short term, and there is a clear need to continue prioritizing natural salt marshes conservation. We also find a number of reasons for the success or failure of planted transplants. Thus, site and species selection, and the application of new planted techniques are important factors to facilitate the establishment of vegetation and avoid project failure. Additionally, our

results show that restoration age are strong predictors of some ecological effects, and surface elevation and sediment characteristics affect the ecological effects of salt marsh restoration. Thus, our analyses provide an opportunity to increase the predictability of restoration outcomes and yield more effective and informed restoration decision-making.

## Methods
### Literature selection
To compile the database, we searched Web of Science Core Collection (1900-2020) in March 2021, using the following search item: TS = (saltmarsh* OR salt marsh*) AND TS = (planting OR plantation OR planted OR reveget* OR transplant* OR restor* OR rehab* OR living shoreline*), resulting in 5,493 publications. Additionally, we searched the bibliography of the papers included from the database search and several previous literature reviews[17,37,40,64], resulting in the inclusion of 110 other published and unpublished studies. In total 5,603 publications were identified.

To include the widest possible range of studies, the following selection criteria were applied: (1) coastal restoration projects or experiments involving the transplanting or seeding of salt marsh vegetation rather than the natural regeneration of salt marsh vegetation; (2) studies conducted in the field, instead of in laboratories, microcosms, tanks, greenhouses or pots; (3) studies with specific location and given planting species. Finally, a total of 210 studies were retained. A list of the publications considered in this systematic review

with meta-analysis is given in Supplementary Data 2. The literature review was conducted according to the Preferred Reporting Items for Systematic Reviews and Meta-Analyses (PRISMA) guidelines[70]. A PRISMA flow diagram showing the literature selection procedure is given in Supplementary Fig. 1. We did not submit the registration form and a review protocol to PROSPERO, because this study is not a human study or animal study relevant to human health.

## Data extraction

For each retained publication, we recorded the following variables: author(s), published year, study location, latitude, longitude, planted age (in years), planted species, planted season, propagule types, survival and factors affecting plant survival and growth. Latitude and longitude data were obtained by locating the study site on Google Earth, when the information is not available in the original text. Data from original figures were extracted using the online tool WebPlotDigitizer (https://apps.automeris.io/wpd/). At the end of the selection, 188 study sites reported in 210 studies were found. An overview map of the worldwide locations of planting in salt marshes is provided in Fig. 1a. Overall 1,038 survival data from 45 plant species reported in 47 studies were found (Fig. 2a, b).

To examine how abiotic-biotic factors affect plant survival and growth, we extracted data on different plant response variables in control and experimental treatments by collecting them from text, tables or digitizing figures. Studies examining the effects of plant sources, fertilization, plant density, plant richness, plant spacing, patch size, and protection on plant survival and growth were retained (Fig. 2f). For plant demographic and growth response, we considered survival, density, height, coverage, aboveground biomass, belowground biomass, and expansion. Studies reporting mean values of the data with sample sizes and some measure of variance (e.g., standard deviations/errors) in both control and experimental treatments were retained. We also compared the survival of the transplanted material with different propagule types, plant seasons, wave forces and herbivory (Fig. 2b–e). It is well known that plants widely differ in salinity, moisture and inundation tolerance[71,72]. The experimental transplant of salt marsh plants across a natural gradient identified different tolerances of physical stress associated with elevation, salinity, soil moisture and tidal inundation[71,72]. Accordingly, we didn't include those studies to calculate the effect sizes in the meta-analysis. Other plant response variables and influencing factors were too few for a meaningful synthesis and were excluded. In total, 762 pairs of observations reported in 41 studies were retained to examine how abiotic-biotic factors affect plant survival and growth (Fig. 2b–f).

To quantify the impacts of planting on ecological outcomes, we restricted our analysis to those studies that compared planted salt marshes with natural or degraded wetlands within the same assessment. Natural wetlands are generally undisturbed areas that have not been affected by severe disturbances, and salt marsh with no evidence of active degradation. Degraded wetlands include abandoned reclamation areas, unvegetated areas and severely disturbed areas previously inhabited by salt marshes that have been affected by natural processes or human activities, such as land reclamation, oil contamination, erosion, high salinity and so on. The unvegetated areas include bare flats caused by the historical loss of salt marshes, or unvegetated mudflat sites resulting from a local dieback or erosion event[73,74]. In this study, restored wetlands refer to planting or revegetation either in areas that were previously salt marsh vegetation but lost or degraded, or new areas that are biophysically suitable for salt marsh vegetation growth or colonization.

Overall, a total of 62 ecological effects variables related to climate change mitigation and biodiversity conservation (Fig. 3) and 16 environmental variables (Fig. 5) were used in this study. These 62 ecological variables were categorized into 6 aggregate categories, namely shoreline protection, primary productivity, soil carbon storage,

greenhouse gas (GHG) fluxes, biodiversity conservation and fishery production. Fishery species are composed of three main groups: fishes, Crustaceans and Mollusca[54]. Fishery species in this study represents the aggregated fishery species without specifying groups (fishes, Crustaceans and Mollusca). Some shoreline protection and soil carbon sequestration variables, such as sediment deposition rate, SOC accumulation rate, SOC stocks, SOC density and SOC content, were not reported or directly available from the publications. We divided the accretion/elevation change, sediment deposition yield and SOC accumulation by the measurement duration (in years) to obtain the averaged rates of accretion/elevation change, sediment deposition rate and SOC accumulation rate[7]. In some studies, SOC content was derived from the measurement of soil organic matter concentration (SOM) by dividing a factor of 1.724[40]. SOC density was calculated using the product of SOC content, soil bulk density and soil depth[75].

To include the widest possible range of data, these studies that had at least one pair of data points of 78 variables comparing planted marshes versus adjacent natural or degraded wetlands were retained. Overall, 9756 pairs of observations reported in 116 studies were retained. Of these, 6377 pairs of observations from 102 publications were variables compared with natural wetlands, and 3379 pairs of observations from 58 publications were variables compared with degraded wetlands. To estimate the effect sizes of planting, studies reporting mean values of the data with sample sizes and standard deviations/errors were retained. Overall, 6570 pairs of observations reported in 115 studies were included. Of these, 4449 pairs of observations from 101 publications compared variables with natural wetlands, and 2121 pairs of observations from 57 publications compared variables with degraded wetlands (Figs. 3 and 5). For each retained publication, we extracted the mean, statistical variation (i.e. standard error, standard deviation) and sample size for each variable at sites where planting was applied and natural or degraded sites from the main text, tables, or by digitizing figures in the articles. Finally, 1038 survival data and 10,518 pairs of observations were included in this systematic review.

## Meta-analysis

To estimate effect sizes of abiotic-biotic factors on each of the above plant demographic and growth variables, we used log response ratio (ln RR) and associated variance to quantify the effect size of abiotic-biotic factors on plant survival, density, height, coverage, aboveground biomass and belowground biomass. Due to the fact that ln RR does not accept non-positive data, the effect sizes of factors on plant expansion were calculated using Hedges' $g*$[76]. Similarly, we calculated ln RR and associated variance as a measure of effect size of restoration on primary productivity, soil carbon storage, biodiversity conservation and fishery production, Hedges' $g*$ was used as a measure of effect size on shoreline protection, GHG fluxes and overall outcomes. Random-effects models were used to estimate the mean effect sizes and 95% confidence intervals (CIs) on each plant demographic and growth variables, as well as on ecological outcomes variables. We calculated the weighted effect size (RR' and $g*$) for the overall restoration outcomes and each type of function. For both ln RR and Hedges' $g*$ effect sizes, mean effect sizes are considered significant if their 95% CIs do not intersect with zero. In these analyses, heterogeneity across studies was estimated using the Cochran's Q statistic (Qt) based on the $\chi^2$ test[76].

To examine if effect sizes of ecological effects increase with planted age, we analysed the relationship between effect size and planted age using meta-regression. In these meta-regression analyses, we used the $Q_M$ statistic to assess the amount of heterogeneity explained by the meta-regression model. Additionally, we examined funnel plot asymmetry quantitatively using the "trimfill" method and estimated Rosenthal's fail-safe number for each effect size metrics to check for the potential influence of publication bias on our results.

These supplementary analyses and results are detailed in Supplementary Fig. 7–10 and Supplementary Table 6–10. The procedures used in the above meta-analyses followed the guidelines of biological meta-analyses[76], and were conducted using R version 4.1.1 and its "metafor" package.

## Reporting summary

Further information on research design is available in the Nature Portfolio Reporting Summary linked to this article.

## Data availability

All data supporting the findings of this study and used to produce the figures have been deposited to the online repository, Figshare[77]. Source data are provided with this paper.

## Code availability

The R code generated for the current study has been deposited to the online repository, Figshare[77].

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

## Acknowledgements

This work is financially supported by the Key Project of National Natural Science Foundation of China (U2243208 and 42330705), National Natural Science Foundation of China (42306187, 42176202 and 32271601), Science & Technology Fundamental Resources Investigation Program (2022FY100304), the National Key Research and Development Program of China (2022YFF1301001-04), the Innovation Group Project of

Southern Marine Science and Engineering Guangdong Laboratory (Zhuhai) (311021004), and the Guangdong Provincial Department of Science and Technology (2019ZT08G090). S.F. is supported by NSF grants DEB-1832221 to the Virginia Coast Reserve Long Term Ecological Research project and OCE-2224608 to the Plum Island Ecosystems Long Term Ecological Research project.

## Author contributions

All authors contributed intellectual input and assistance to this study. Z.Z.L. and B.S.C. designed the study; Z.Z.L. collected, analyzed the data and created the figures; Z.Z.L. wrote the first draft of the manuscript; Z.Z.L., S.F., Q.H., O.G., Z.H. and B.S.C. discussed the results; all authors contributed substantially to revisions.

## Competing interests

The authors declare no competing interests.
