## [Peer Review File · Nature Communications]

Reviewers' Comments:

Reviewer #1:

Remarks to the Author:

This study presents results from a meta-analysis of the success (survival and growth) and ecological benefits of salt marsh plantings as compared to natural and degraded marshes globally. The study is timely given the increasing interest in restoration and nature-based solutions to climate change and other "wicked" socio-ecological problems we are facing globally. However, I think this manuscript would require significant revisions and perhaps additional analyses before it would be a meaningful contribution. I have concerns about the data analyses, particularly the sample sizes and study exclusion criteria, as well as the interpretation of the results. My comments and concerns, both major and minor, are outlined by line below.

Line 90-91: Incomplete sentence or change "the" to "a" if referring to meta-analysis techniques in general?

Line 110: Were the plantings in many studies intended to be experimental? If so, wouldn't survival outcomes be expected to be low if treatments predicted to perform poorly are included? Would it be more helpful to report survival rates for the treatments hypothesized to have the highest survival as compared to the lowest survival in each study (if the study intentionally manipulated conditions to better understand planting survival)? Restoration monitoring studies could be reported separately from studies with experimental manipulations of plantings.

Line 167: But carbon sequestration is not just about soil organic carbon accumulation rates, it is about the total already accumulated. Restored marshes cannot make up for the loss of natural marshes in terms of losses of carbon to the atmosphere. This should be reworded to acknowledge that restored marshes are not equivalent to natural marshes in the context of carbon sequestration.

Line 168: Improper use of the word "Similarly", the two statements are in contrast to one another. Also Figure 3 suggests that there is no statistically significant difference between natural marsh and restored marshes in terms of fishery production, I don't even see fishery production on the graph just species richness of fishery and fish species.

Line 169- 173: Some of the results reported are contradictory. The authors first say that fish production is lower in restored marshes as compared to natural marshes and then say that planted marshes support more fish than natural marshes, which is it?

Line 177: It would be helpful to have a definition of what was considered "restored" versus "natural" versus "degraded" up front so that it is easier to interpret the results.

Lines 184- 188: Again, lack of clarity in the responses, how are "individual functions in biodiversity and fishery production different than species richness and diversity of fish? Contradictory results.

Line 189- 193: It would be helpful to have the numbers of studies used in each comparison included in (N= X).

Line 200: I do not think the data shown in Figure 3 supports the claim that high-survival species provided greater ecological benefits than low-survival species. I would say the comparisons are mixed at best and given the number of studies used in these comparisons are very small (N=1, N=2 in many cases for low survival species), I am not sure I would include this comparison in the study. Are species classified as high survival or low survival based on actual survival in the studies for which metrics are compared or are the species simply being categorized by average survivorship across all studies? If a species had really low survival in the 1 or 2 studies used to compare ecological metrics to a species with very high survival, then one would assume the ecological benefits would be lower in the low survival group. However, if comparisons are made between studies with approximately equal survival, then one could make comparisons across species in terms of ecological benefits.

Lines 203-207: I do not agree with interpretation of the SOC stocks and content comparisons based on the graphs shown. SOC content and stocks were lower in restored when compared to natural wetlands in Figure 3 and Figure 4. The number of studies of low survival species is too low (N=2 or 1) to make any meaningful comparison for SOC.

Lines 227: -233: the interpretation of the results here is not consistent with the reported results previously. I suggest the authors review the reported results and revise accordingly.

Lines 244: I thought studies evaluating salinity were excluded?

Lines 255-257: What do you mean by "significant: or "Huge potential"? These are grandiose/vague terms.

Lines 260-261: Vertical accretion and elevation change would be expected to happen at a higher rate at restored sites when compared to natural sites because the restored sites are likely starting at a lower elevation and have less sediment trapping capability without existing vegetation (as is stated later in the discussion). I don't think it is fair to say that these changes are not related to age, but instead that there were enough long-term studies to determine if/when rates of elevation change and sediment accretion level off and become more similar to natural marshes.

Lines 283-284: What data do you show to support this statement? I don't see a figure referenced here.

Lines 362-365: I don't understand why these studies were excluded. If the goal was to examine how abiotic-biotic factors affect plant survival and growth, excluding the most critical abiotic factors does not seem like a sound approach. Additionally, if the authors want to draw conclusions about the role of plantings in the context of climate change, excluding studies that looked at tidal inundation, salinity, and elevation does not make sense. Additionally, these are factors that are often manipulated in conjunction with other abiotic or biotic factors, so this would seem to severely limit the pool of studies considered.

Lines 370-372: The strict criteria used to select studies has severely limited the scope of this meta-analysis, 33 studies is a fairly small sample size for a meta-analysis. Including studies that considered the abiotic factors initially excluded would likely increase this sample size and improve this meta-analysis.

Line 377: Why is an unvegetated tidal flat considered a degraded wetland? I would assume a tidal flat would only be considered degraded if it was clearly denoted as previously vegetated and something caused the vegetation to die off. Please clarify.

Figure 3: The groupings of response metrics are confusing, why is there a "species richness of fishes" and a "species richness of fishery species"?

Reviewer #2:

Remarks to the Author:

This study assembled a global database, encompassing 21,012 observations reported in 210 studies, to examine the drivers and impacts of salt marsh planting. Meta-analysis was used for the data analysis and results got. The study found that only half of plantings survived, with survival predicted mainly by species characteristics and abiotic and biotic filters. Planting enhanced carbon sequestration, biodiversity, and shoreline protection among other ecosystem services as compared with degraded wetlands. However, the ecosystem services of planted marshes, except for shoreline protection, have not yet fully recovered compared to natural wetlands. Planting performance tended to increase with time since planting and for species with high survival rates. The results suggested that salt marsh planting is a promising strategy for biodiversity conservation and climate change mitigation and have important implications for designing coastal restoration strategies.

The aims to analysis the survival of saltmarsh planting projects and the comparison between

natural and degraded wetlands. The study is a guide to the success of saltmarsh planting and informed restoration decision-making. The methodology sound. However, the writing and analysis of results still needs significant improvement.

1. The title does not match the topic of the manuscript. The key questions of the manuscript was to analysis the survival and their environmental factors, and the ecosystem services of saltmarsh by comparing with degraded saltmarsh. The title went a little far from these theme.

2. The new findings of the the meta-analysis were still not clearly stated, and need to distinguish from the common sense. The data from the analysis should be included in the results and in abstracts, which would be very important for the reader to quantify the results.

3. The results of species and age of planting are very important for the success of saltmarsh restoration. That is a key contribution to the restoration works. The environmental factors that the authors analyzed are very helpful the restoration practices, e.g. species selection, time, aging, environmental stress... they would definitely contribute the saltmarsh planting projects. I would suggest the authors narrowed the focus and strengthen this instated of on "climate change mitigation". For the climate change mitigation, the greenhouse gas flux, the C gain by plants and the C flux should be also address in the study. This part is too weak in the main text.

Line numbers in our responses refer to the revised manuscript without tracked changes.

Response to Reviewer #1

This study presents results from a meta-analysis of the success (survival and growth) and ecological benefits of salt marsh plantings as compared to natural and degraded marshes globally. The study is timely given the increasing interest in restoration and nature-based solutions to climate change and other “wicked” socio-ecological problems we are facing globally. However, I think this manuscript would require significant revisions and perhaps additional analyses before it would be a meaningful contribution. I have concerns about the data analyses, particularly the sample sizes and study exclusion criteria, as well as the interpretation of the results. My comments and concerns, both major and minor, are outlined by line below.

Response: Many thanks to the reviewer’s positive comments on our work. We have revised the manuscript following the constructive and helpful suggestions from the reviewer.

Line 90-91: Incomplete sentence or change “the” to “a” if referring to meta-analysis techniques in general?

Response: Thank you for your suggestion. We have changed it to “For each variable related to shoreline protection, primary production, soil carbon storage, greenhouse gas (GHG) fluxes, biodiversity conservation and fishery production, we conducted a meta-analysis to quantify both log response ratios and Hedges’ g^* effect sizes, as a means of providing a quantitative estimate of salt marsh planting performance.” (Lines 92-96).

Line 110: Were the plantings in many studies intended to be experimental? If so, wouldn’t survival outcomes be expected to be low if treatments predicted to perform poorly are included? Would it be more helpful to report survival rates for the treatments hypothesized to have the highest survival as compared to the lowest survival in each study (if the study intentionally manipulated conditions to better understand planting survival)? Restoration monitoring studies could be reported separately from studies with experimental manipulations of plantings.

Response: Thank you for this suggestion. We have separated experimental manipulations of plantings from restoration sites, and found that there was no statistically significant difference between restoration projects and experimental manipulations of plantings (Supplementary Fig. 2) (Lines 114-115).

Actually, we focus on empirical field studies and exclude studies from experiments in

laboratories, microcosms, tanks, greenhouses or pots. Generally, the field experiments tend to adopt standardised protocols for sampling, data collection, and statistical analysis, and using robust experimental designs and methods.

Supplementary Figure 2. Differences in survival rate (\pm SE) between restoration projects and experimental manipulations of plantings. Differences in survival rate were assessed using two-tailed Student t-tests.

Line 167: But carbon sequestration is not just about soil organic carbon accumulation rates, it is about the total already accumulated. Restored marshes cannot make up for the loss of natural marshes in terms of losses of carbon to the atmosphere. This should be reworded to acknowledge that restored marshes are not equivalent to natural marshes in the context of carbon sequestration.

Response: Thank you for this suggestion. In the revised manuscript, we added one sentence in the Results (Line 173-184) to declare this point.

“Planted marshes provide lower levels of primary productivity and soil carbon storage compared with natural wetlands. Belowground biomass (n = 317) and subsurface soil carbon content/density (> 10 cm deep) (n = 99 and 34, respectively) develop more slowly than aboveground biomass (n = 359) and surface soils (0–10 cm deep) (n = 136 and 14, respectively) (Fig. 3a). However, the level of soil organic carbon (SOC) accumulation rate was not different between restored (64.26 ± 7.54 g/m²/year) and natural (64.72 ± 9.31 g/m²/year) marshes

(n = 18) (Table 1). In addition, there were no significant differences in CO₂ (n = 86, Hedges' $g^* = 0.03$, 95% CIs = -0.10 to 0.17) and N₂O (n = 70, Hedges' $g^* = 0.12$, 95% CIs = -0.03 to 0.27) fluxes between restored and natural marshes, and restored marshes have significantly lower CH₄ flux (n = 70, Hedges' $g^* = -0.21$, 95% CIs = -0.38 to -0.03) than their natural sites (Fig. 3a). These suggested that restored marshes have the potential to equal or exceed the carbon sequestration capacity of the natural marshes.”

Line 168: Improper use of the word “Similarly”, the two statements are in contrast to one another. Also Figure 3 suggests that there is no statistically significant difference between natural marsh and restored marshes in terms of fishery production, I don't even see fishery production on the graph just species richness of fishery and fish species.

Response: Thank you for this suggestion. We have revised these sentences to make our point clearer. See the revised version in lines 185-191:

“Furthermore, the levels of biodiversity of vegetation and macrobenthos are much lower for restored marshes compared to natural marshes. For instance, restored marshes have lower levels of species richness for vegetation (n = 26, $RR' = -0.41$, 95% CIs = -0.57 to -0.24) and macrobenthos (n = 64, $RR' = -0.23$, 95% CIs = -0.34 to -0.12), and less abundance and biomass of macrobenthos (428 and 62, respectively) than natural marshes. Similarly, planted sites have less size, biomass and abundance of fishery species than natural marshes (84, 64 and 746, respectively) (Fig. 3a and Table 1).”

Line 169- 173: Some of the results reported are contradictory. The authors first say that fish production is lower in restored marshes as compared to natural marshes and then say that planted marshes support more fish than natural marshes, which is it?

Response: We apologize for the confusion. We have revised these sentences to make our point clearer. See the revised version in lines 189-194:

“Similarly, planted sites have less size, biomass and abundance of fishery species than natural marshes (84, 64 and 746, respectively) (Fig. 3a and Table 1). However, this is not evident for all individual functions, for example, planted marshes tended to support higher abundance of birds (n = 27, $RR' = 0.64$, 95% CIs = 0.08 to 1.20) and catch rate of fishery species (n = 147, $RR' = 0.50$, 95% CIs = 0.34 to 0.65) than nearby natural marshes (Fig. 3a).”

Line 177: It would be helpful to have a definition of what was considered “restored” versus

“natural” versus “degraded” up front so that it is easier to interpret the results.

Response: Thank you for this suggestion. We have added definition in the Material and Methods in lines 398-406:

“Natural wetlands are generally undisturbed areas that have not been affected by severe disturbances, and salt marsh with no evidence of active degradation. Degraded wetlands include abandoned reclamation areas, unvegetated areas and severely disturbed areas previously inhabited by salt marshes that have been affected by natural processes or human activities, such as land reclamation, oil contamination, erosion, high salinity and so on. The unvegetated areas include bare flats caused by the historical loss of salt marshes, or unvegetated mudflat sites resulting from a local dieback or erosion event^{72,73}. In this study, restored wetlands refer to planting or revegetation either in areas that were previously salt marsh vegetation but lost or degraded, or new areas that are biophysically suitable for salt marsh vegetation growth or colonization.”

Lines 184- 188: Again, lack of clarity in the responses, how are “individual functions in biodiversity and fishery production different than species richness and diversity of fish? Contradictory results.

Response: We apologize for the confusion. We have revised these sentences to make our point clearer. See the revised version in lines 205-201:

“Furthermore, planted marshes have higher species richness of vegetation, macrobenthos and fishery species than degraded marshes (n = 20, 22 and 17, respectively). The size, biomass, abundance and catch rate of fishery species are much higher for restored marshes compared with degraded marshes (n = 45, 28, 438 and 68, respectively). However, there was no major difference for other functions such as abundance and biomass of macrobenthos (n = 405 and 63, respectively) (Fig. 3b and Table 1).”

Line 189- 193: It would be helpful to have the numbers of studies used in each comparison included in (N= X).

Response: Thank you for this suggestion. We have added the numbers of studies in each comparison.

Line 200: I do not think the data shown in Figure 3 supports the claim that high-survival species provided greater ecological benefits than low-survival species. I would say the comparisons are mixed at best and given the number of studies used in these comparisons are very small

(N=1, N=2 in many cases for low survival species), I am not sure I would include this comparison in the study. Are species classified as high survival or low survival based on actual survival in the studies for which metrics are compared or are the species simply being categorized by average survivorship across all studies? If a species had really low survival in the 1 or 2 studies used to compare ecological metrics to a species with very high survival, then one would assume the ecological benefits would be lower in the low survival group. However, if comparisons are made between studies with approximately equal survival, then one could make comparisons across species in terms of ecological benefits.

Response: Thank you for this suggestion. We have removed the analysis of the effect of restoration species because the number of studies used in these comparisons is very small.

Lines 203-207: I do not agree with interpretation of the SOC stocks and content comparisons based on the graphs shown. SOC content and stocks were lower in restored when compared to natural wetlands in Figure 3 and Figure 4. The number of studies of low survival species is too low (N=2 or 1) to make any meaningful comparison for SOC.

Response: Thank you for this suggestion. We have removed the analysis of the effect of restoration species because the number of studies used in these comparisons is very small.

Lines 227: -233: the interpretation of the results here is not consistent with the reported results previously. I suggest the authors review the reported results and revise accordingly.

Response: Thank you for this suggestion. We have revised these sentences. See the revised version in lines 248-258:

“This systematic review suggests that salt marsh planting is a promising strategy for biodiversity conservation and climate change mitigation and adaptation. Specifically, we find that mean survival is 53% for transplants, limited by intrinsic species characteristics and a series of abiotic and biotic filters. However, careful design of sites, species selection, and novel planted technologies can facilitate the survival and establishment of planted organisms. Additionally, our study shows that planting enhanced carbon sequestration, biodiversity, and shoreline protection as compared with degraded wetlands. Compared with natural wetlands, however, the ecosystem services of planted marshes, except for shoreline protection, have not yet fully recovered. Fortunately, most effects of planting on carbon storage and biodiversity conservation increased with planted age. Thus, salt marsh planting contributes to the implementation of global agendas for climate change adaptation and mitigation, biodiversity conservation and sustainable development.”

Lines 244: I thought studies evaluating salinity were excluded.

Response: Thank you for this suggestion. We have removed “hypersalinity”.

Lines 255-257: What do you mean by “significant: or “Huge potential”? These are grandiose/vague terms.

Response: Thank you for this suggestion. We have removed “significant” and “Huge potential”, and revised these sentences. See the revised version in lines 274-2279:

“For instance, the effect size of primary productivity and soil carbon storage compared with natural wetlands increased with planted age (Fig. 4a and b), indicating that planting in salt marshes restored the capacity of the ecosystem to store carbon and help rebuild the lost carbon sink. The effect sizes of macrobenthos species richness and abundance are also strongly related to planted age, which provides evidence of a clear benefit of planting for biodiversity conservation (Fig. 4a and b).”

Lines 260-261: Vertical accretion and elevation change would be expected to happen at a higher rate at restored sites when compared to natural sites because the restored sites are likely starting at a lower elevation and have less sediment trapping capability without existing vegetation (as is stated later in the discussion). I don’t think it is fair to say that these changes are not related to age, but instead that there were enough long-term studies to determine if/when rates of elevation change and sediment accretion level off and become more similar to natural marshes.

Response: Thank you for this suggestion. We have deleted the discussion of vertical accretion and elevation change, and revised these sentences. See the revised version in lines 281-284:

“However, several restoration benefits are not sensitive to planted age, such as species richness of overall fishery species and diversity of macrobenthos. These processes may be more sensitive to other factors such as hydrographic conditions, food resources and refuge structures among habitats^{45,53,54}.”

Lines 283-284: What data do you show to support this statement? I don’t see a figure referenced here.

Response: This result can be seen in Fig. 5b and Supplementary Table 4. See the revised version in lines 304-306:

“Surface elevation and soil grain size were comparable to those in degraded wetlands (Fig. 5b; Supplementary Table 4), indicating that restored marshes had similar hydrologic and

geomorphic regimes.”

Lines 362-365: I don't understand why these studies were excluded. If the goal was to examine how abiotic-biotic factors affect plant survival and growth, excluding the most critical abiotic factors does not seem like a sound approach. Additionally, if the authors want to draw conclusions about the role of plantings in the context of climate change, excluding studies that looked at tidal inundation, salinity, and elevation does not make sense. Additionally, these are factors that are often manipulated in conjunction with other abiotic or biotic factors, so this would seem to severely limit the pool of studies considered.

Response: We apologize for the confusion. We didn't delete the studies evaluating elevation, salinity, moisture and tidal inundation. We just didn't include those studies to calculate the effect sizes in the meta-analysis (Fig. 2f). Across the entire dataset reported in 210 studies, a series of abiotic and biotic factors, including elevation, salinity, moisture and tidal inundation, are critical determinants of transplant's survival and growth (Fig. 2a).

Figure 2. Constraints on planting establishment and the effect size of plant performances.

a, Planting establishment is influenced by ecological filters and species characteristics

including a series of abiotic and biotic factors. Sankey diagram in which the thickness of the lines between the left and right columns represents the number of studies between the factors and plant genus they involved. Number of studies is indicated with n. b-e, Plant survival rates under different propagule types (b), planting season (c), wave force (d) and herbivory (e). Each line represents the difference in survival rate in a planted with seed and seedlings pair (b), planted in spring and autumn season pair (c), planted in the wave-exposed site and sheltered site pair (d), plant in an herbivory and control treatment with grazing exclusion pair (e).

However, it is well known that plants widely differ in salinity, moisture and inundation tolerance. For example, Burchett et al. (1999) conducted a 3-year saltmarsh transplantation project to compare the survival and growth of six plant species across a natural tidal gradient. Final field survival rates are presented in Fig. S1. *Sarcocornia quinqueflora* showed significantly higher survival rates up the whole tidal gradient than *Suaeda australis* and *Sporobolus virginicus*. *Suaeda australis* and *Sporobolus virginicus* needed a higher level of soil water, which was supplied by more frequent tidal inundation at the lower elevations, and by a slightly greater freshwater input from the bank behind at the upper edges of the site (Fig S1a). *Halosarcia pergranulata* showed the highest survival rates, sustained up the entire tidal gradient. However, *Lampranthus tegens* survived and flourished only at the uppermost elevations of the ramp, and *Wilsonia backhousei* had very low field survival rates at both the lowest and highest elevations (Fig S1b).

Fig. S1. Field survival rates after 2 years of growth of (a) the dominant species *Sarcocornia quinqueflora* (Sarc), *Suaeda australis* (Suaed) and *Sporobolus virginicus* (Spor); and (b) the rare species *Halosarcia pergranulata* (Halo), *Lampranthus tegens* (Lamp) and *Wilsonia backhousei* (Wil), at successive 10 cm elevations up experimental ramp (elevations relative to mean high water).

In another case, Konisky et al. (2004) conducted an experiment to transplant six New England plant species across a natural physical gradient of three tidal flooding and three salinity regimes. They found that the salt-tolerant species, *Spartina alterniflora*, *Spartina patens*, and *Juncus gerardii*, showed greater overall survival than the brackish species, *Phragmites australis*, *Typha angustifolia*, and *Lythrum salicaria*, since study locations were mesohaline or polyhaline (Fig. S2). Biomass results were also highly variable across species (Fig. S2). These results, combined with survival data, strongly suggested that species were impacted differentially at gradient locations. In some locations, all individuals of a species died; in others, plants survived but grew poorly or grew seemingly unaffected by stress.

Fig. S2. Species final aboveground (solid) and belowground (hashed) biomass (mean + SE). Y-axis scale 0–80 g for *S. alterniflora* and *Typha*, 0–40 g for others. Number of surviving transplants (out of 14 initial) in parentheses.

Therefore, the experimental transplant of common salt marsh plants across a natural gradient of flood and salinity regimes identified different tolerances of physical stress associated with saltwater flooding. To clarify this point, we have rewritten these sentences. See the revised version in lines 389-393:

“It is well known that plants widely differ in salinity, moisture and inundation tolerance^{70,71}. The experimental transplant of salt marsh plants across a natural gradient identified different tolerances of physical stress associated with elevation, salinity, moisture and tidal

inundation^{70,71}. Accordingly, we didn't include those studies to calculate the effect sizes in the meta-analysis.”

Burchett, M. D., Allen, C., Pulkownik, A. & MacFarlane, G. Rehabilitation of saline wetland, Olympics 2000 site, Sydney (Australia). II: saltmarsh transplantation trials and application. *Mar. Pollut. Bull.* **37**, 526-534 (1999).

Konisky, R. A., and D. M. Burdick. Effects of stressors on invasive and halophytic plants of New England salt marshes: a framework for predicting response to tidal restoration. *Wetlands* **24**: 434–447 (2004).

Lines 370-372: The strict criteria used to select studies has severely limited the scope of this meta-analysis, 33 studies is a fairly small sample size for a meta-analysis. Including studies that considered the abiotic factors initially excluded would likely increase this sample size and improve this meta-analysis.

Response: Thank you for this suggestion. Based on our criteria, a total of 210 articles were used in our study. For constraints on the fitness of outplants, we have included these studies compared the survival with different propagule types, plant seasons, wave forces and herbivory. Thus, 41 publications, which yielded 762 sets of pairwise comparisons, were retained to examine how abiotic-biotic factors affect plant survival and growth (Fig. 2 b-f).

Moreover, a dataset of 41 publications is an adequate sample size for a meta-analysis. For example, a global meta-analysis examining how insects promote crop pollination was based on 23 studies (Woodcock et al., 2019); Hua et al. (2024) provided a large-scale test of the biodiversity impacts of agricultural deforestation using a global database reported in 44 primary studies; Větrovský et al. (2019) and Gardner et al. (2019) conducted a global meta-analysis using the final dataset consisted of data derived from 36 and 43 studies, respectively. To clarify this point, thus, we have rewritten these sentences. See the revised version in lines **394-395**:

“In total, 762 pairs of observations reported in 41 studies were retained to examine how abiotic-biotic factors affect plant survival and growth (Fig. 2 b-f).”

Woodcock, B.A., Garratt, M.P.D., Powney, G.D. et al. Meta-analysis reveals that pollinator functional diversity and abundance enhance crop pollination and yield. *Nat Commun* **10**, 1481 (2019). <https://doi.org/10.1038/s41467-019-09393-6>

Hua, F., Wang, W., Nakagawa, S. et al. Ecological filtering shapes the impacts of agricultural

deforestation on biodiversity. *Nat Ecol Evol* (2024). <https://doi.org/10.1038/s41559-023-02280-w>

Větrovský, T., Kohout, P., Kopecký, M. et al. A meta-analysis of global fungal distribution reveals climate-driven patterns. *Nat Commun* 10, 5142 (2019). <https://doi.org/10.1038/s41467-019-13164-8>

Gardner, C.J., Bicknell, J.E., Baldwin-Cantello, W. et al. Quantifying the impacts of defaunation on natural forest regeneration in a global meta-analysis. *Nat Commun* 10, 4590 (2019). <https://doi.org/10.1038/s41467-019-12539-1>

Line 377: Why is an unvegetated tidal flat considered a degraded wetland? I would assume a tidal flat would only be considered degraded if it was clearly denoted as previously vegetated and something caused the vegetation to die off. Please clarify.

Response: We apologize for the confusion. The Reviewer is correct. Degraded wetlands were clearly indicated as such in the original individual studies we used to extract the data for this meta-analysis. We used “unvegetated tidal flats” to represent bare flats caused by the historical loss of salt marshes, or unvegetated mudflat sites resulting from a local dieback event.

To clarify this point, thus, we have rewritten these sentences. See the revised version in lines 399-404:

“Degraded wetlands include abandoned reclamation areas, unvegetated areas and severely disturbed areas previously inhabited by salt marshes that have been affected by natural processes or human activities, such as land reclamation, oil contamination, erosion, high salinity and so on. The unvegetated areas include bare flats caused by the historical loss of salt marshes, or unvegetated mudflat sites resulting from a local dieback or erosion event^{72,73}.”

Cain, J. L., & Cohen, R. A. Using sediment alginate amendment as a tool in the restoration of *Spartina alterniflora* marsh. *Wetl. Ecol. Manage.* **22**, 439-449 (2014).

Curado, G., Figueroa, E., Castillo, J. M. Vertical sediment dynamics in *Spartina maritima* restored, non-restored and preserved marshes. *Ecol. Eng.* **47**, 30–35 (2012).

Figure 3: The groupings of response metrics are confusing, why is there a “species richness of fishes” and a “species richness of fishery species”?

Response: We apologize for the confusion. Fishery species in salt marshes are composed of three main groups: fishes, Crustaceans and Mollusca. Of these fishery species, some common fishes include mullets and goby, some common Crustaceans include crabs and shrimp, and

some common Mollusca include mussels, snails and clams. Species richness of fishery species represents the total number of overall fishery species without specifying the fishes, crustaceans and Mollusca, while species richness of fishes represents the total number of fishes species.

To clarify this point, we have added some sentences in the Materials and Methods. See the revised version in lines **411-413**:

“Fishery species are composed of three main groups: fishes, Crustaceans and Mollusca⁵⁴. Fishery species in this study represents the aggregated fishery species without specifying groups (fishes, crustaceans and Mollusca).”

Minello, T. J. & J. W. Webb, Jr. Use of natural and created *Spartina alterniflora* salt marshes by fishery species and other aquatic fauna in Galveston Bay, Texas, USA. *Mar. Ecol. Prog. Ser.* **151**, 165–179 (1997).

Response to Reviewer #2

This study assembled a global database, encompassing 21,012 observations reported in 210 studies, to examine the drivers and impacts of salt marsh planting. Meta-analysis was used for the data analysis and results got. The study found that only half of plantings survived, with survival predicted mainly by species characteristics and abiotic and biotic filters. Planting enhanced carbon sequestration, biodiversity, and shoreline protection among other ecosystem services as compared with degraded wetlands. However, the ecosystem services of planted marshes, except for shoreline protection, have not yet fully recovered compared to natural wetlands. Planting performance tended to increase with time since planting and for species with high survival rates. The results suggested that salt marsh planting is a promising strategy for biodiversity conservation and climate change mitigation and have important implications for designing coastal restoration strategies.

The aims to analysis the survival of saltmarsh planting projects and the comparison between natural and degraded wetlands. The study is a guide to the success of saltmarsh planting and informed restoration decision-making. The methodology sound. However, the writing and analysis of results still needs significant improvement.

Response: We are grateful for this careful review and insightful comments. We have carefully considered all comments, which helped improving the quality of the manuscript.

1. The title does not match the topic of the manuscript. The key questions of the manuscript was to analysis the survival and their environmental factors, and the ecosystem services of saltmarsh by comparing with degraded saltmarsh. The title went a little far from these theme.

Response: Thank you for your suggestion. We have changed it to “Drivers of success in salt marshes restoration through planting and implications for ecosystem services”.

2. The new findings of the meta-analysis were still not clearly stated, and need to distinguish from the common sense. The data from the analysis should be included in the results and in abstracts, which would be very important for the reader to quantify the results.

Response: Thank you for your comment. Originally, we avoided adding quantitative estimates in the narrative of the abstract and results to improve readability, as all the quantitative estimates were visible in the Figures and Tables. However, following the suggestion of the Reviewer we added many quantitative estimates throughout the Abstract and Results. However, we should point out that the abstracts and main text (not including Abstract, Methods,

References and Figure legends) should be limited to 200 and 5,000 words, so we cannot expand the results in great length. The abstract was revised to be clearer. We have to note that due to the strict word count, we cannot include a more extended version of the results in the abstract

3. The results of species and age of planting are very important for the success of saltmarsh restoration. That is a key contribution to the restoration works. The environmental factors that the authors analyzed are very helpful the restoration practices, e.g. species selection, time, aging, environmental stress... they would definitely contribute the saltmarsh planting projects.

Response: Thank you for your comment. The Reviewer is correct. Unfortunately, for some functions there is a general lack of studies. This is a feature of the literature that is beyond the control of the authors. Following the comments of the Reviewer #1, we have removed the analysis of the effect of restoration species because of low sample size. Furthermore, we point out the environmental factors affect the ecological benefits of salt marsh restoration in the Discussion.

In any case, we believe that this is an interesting reflection/discussion point in its own right indicating major knowledge gaps that become particularly pertinent to be addressed, now that we enter the Decade of Restoration as designated by United Nations. We will be happy to revise further if the Reviewer wishes to clarify better this specific point.

I would suggest the authors narrowed the focus and strengthen this instated of on “climate change mitigation”. For the climate change mitigation, the greenhouse gas flux, the C gain by plants and the C flux should be also address in the study. This part is too weak in the main text.

Response: Thank you for your comment. We have added the effects of salt marsh planting on greenhouse gas fluxes (CO₂, CH₄ and N₂O fluxes) in the revised manuscript. In the original manuscript, we calculated the effect sizes of 18 ecological functions related to climate change mitigation, including primary productivity, SOC content, SOC density, SOC stocks and SOC accumulation rate.

Reviewers' Comments:

Reviewer #1:

Remarks to the Author:

The authors have adequately addressed my comments in the revised manuscript.

Reviewer #2:

Remarks to the Author:

The author has made a serious revision according to the previous review comments. The comments on the theme, data analysis, etc. of the manuscript have been revised. The manuscript looks very good now, and I have no further suggestions.

Response to Reviewer #1

The authors have adequately addressed my comments in the revised manuscript.

Response: Thank you very much for your positive and kind comments.

Response to Reviewer #2

The author has made a serious revision according to the previous review comments. The comments on the theme, data analysis, etc. of the manuscript have been revised. The manuscript looks very good now, and I have no further suggestions.

Response: Thank you very much for your positive and kind comments.